# Nanomotion technology for testing azithromycin susceptibility of *Salmonella enterica*

Mariliis Hinnu,[1] Toomas Mets,[1] Ivana Kerkez,[1] Marta Putrinš,[1] Niilo Kaldalu,[1] Gino Cathomen,[2] Marta Pla Verge,[2] Danuta Cichocka,[2] Alexander Sturm,[2] Tanel Tenson,[1] for the ERADIAMR consortium

**ABSTRACT** Invasive salmonellosis caused by *Salmonella enterica* subspecies affects millions of people every year, mostly children from low-income countries, and is associated with a high mortality rate. Azithromycin is used to treat invasive salmonellosis resistant to first-line drugs despite conflicting effective concentrations *in vitro* and achievable serum concentrations *in vivo*. As resistance levels to azithromycin are rising, we demonstrate that nanomotion technology, which is based on measuring changes in bacterial nanoscale movements, can be used for rapid phenotypic testing of *Salmonella*'s susceptibility to azithromycin. Additionally, the use of nanomotion enabled the detection of the bactericidal effect. Nanomotion changes under various culture conditions correlated with susceptibility measured by minimum inhibitory concentration (MIC) determination, colony-forming unit (CFU) counting, and fluorescent reporter-based estimates of intrabacterial azithromycin accumulation. Environmental conditions, both during azithromycin treatment and throughout the recovery period, significantly affect the antibacterial response to azithromycin. Azithromycin susceptibility in *Salmonella* is detectable after only 2 h of treatment. This reflects the quick action of the antibiotic, which could be one of the contributing factors behind the clinical efficacy of azithromycin for *Salmonella* treatment. Our study underscores the critical role of assay conditions, which greatly influenced both azithromycin efficacy and the test results.

**IMPORTANCE** Azithromycin is used as a last-resort antibiotic to treat life-threatening infections caused by *Salmonella enterica*, a high-priority pathogen according to the World Health Organization. Resistance levels to azithromycin are increasing, highlighting the need for rapid susceptibility testing. In this study, we demonstrate that nanomotion technology can detect azithromycin susceptibility in *Salmonella*, suggesting its potential use for rapid resistance detection in clinical settings and its future use with azithromycin. Additionally, the study shows that nanomotion technology can be used for susceptibility and postantibiotic effect testing for various pathogens and antibacterials, including those generally regarded as bacteriostatic.

**KEYWORDS** rapid antimicrobial susceptibility detection, azithromycin, *Salmonella*, diagnostic, nanomotion

Invasive salmonellosis caused by *Salmonella enterica* subspecies is a major threat to human health affecting >20 million people yearly (1, 2). Antimicrobial resistance to traditional drugs, such as beta-lactams and fluoroquinolones, has been reported in all clinically relevant *S. enterica* serovars (1). The macrolide azithromycin (AZI) has been effectively used to treat *Salmonella* infections resistant to other drug classes (1). AZI remains effective *in vivo*, despite recommended doses achieving peak serum concentrations in the range of 0.4 µg/mL (3): 20-fold lower than the minimum inhibitory concentrations (MICs) for most clinical strains (8 µg/mL) (4). Resistance to AZI is increasing (5,

**Peer Reviewers** Mor N. Lurie-Weinberger, Tel Aviv Sourasky Medical Center, Tel Aviv, Israel; Ying-Tsong Chen, National Chung Hsing University, Taichung, Taiwan; Garrett Ellward, University of Florida, Gainesville, Florida, USA

Address correspondence to Mariliis Hinnu, mariliis.hinnu@ut.ee.

G.C., M.P.V., D.C., and A.S. are employees of Resistell AG and declare competing financial interests.

See the funding table on p. 8.

6), underlining the need for rapid susceptibility testing. Nanomotion technology can be used as a rapid phenotypic antimicrobial susceptibility test (AST) (7–11). The technology, which is based on measuring oscillations (nanomotions) caused by metabolically active organisms attached to a nanomechanical sensor (a cantilever), has been described in detail in the referenced articles (7–13). Effective drug concentrations reduce or stop the nanomotions, and the classification into resistant/susceptible categories is based on machine learning (ML) algorithms for specific strain-drug combinations. The ML model is trained on a large set of clinical isolates, based on the standard MIC and nanomotion response at different drug concentrations. The susceptibility phenotype can already be detected 2 h after blood culture positivity (11). The technology has been successfully applied in various bacterial species and two clinical studies (NANO-RAST [13], NCT05002413) and PHENOTECH-1 ([11], NCT05613322) for determining antibiotic susceptibility (beta-lactams and fluoroquinolones) of gram-negative bacteria causing bacteremia and/or sepsis with >90% accuracy.

Prior to this study, nanomotion had not been used to determine susceptibility to AZI or any other macrolide. We recorded nanomotion of *S. enterica* under various experimental conditions affecting its susceptibility to AZI. We used neutral and acidic media, a condition encountered by intracellular *Salmonella* in acidic vacuoles (14), and two different incubation temperatures. In the early stages of development, nanomotion was measured at ambient room temperature (RT). The current setup uses 37°C for all ASTs to mimic physiological conditions in humans and to decrease the time to results (11).

The susceptibility of *S. enterica* serovar Typhimurium SL1344 (15) (wild-type; wt) to AZI was studied using nanomotion and standard culture-based assays at both RT and 37°C. Based on MIC values, *Salmonella* is up to four times more sensitive to AZI at RT compared to 37°C in different growth media (Fig. S1A; Table S1). This is not fully explainable by the differences in growth rates, as different media can have the same MICs at different growth rates or different MICs at similar growth rates (Fig. S1B). Nanomotion was recorded during AZI treatment and subsequent recovery in fresh drug-free medium (Fig. 1 and S2). Before the addition of the antibiotic, nanomotion variance over time increased, indicating the presence and physiological activity of live bacteria on the cantilever. In the untreated sample, the signal continued to increase during the measurement (Fig. 1A). When AZI was added at concentrations at or exceeding the MIC, the nanomotion signal slope decreased (Fig. 1B and C). After the removal of AZI, nanomotion started to increase again in fresh drug-free medium, indicating recovery (Fig. 1B and C; Fig. S2). No recovery was observed after treatment with high AZI concentrations when the experiments were conducted at RT (Fig. 1C, Fig. S2). In all cases, bacteria remained on the cantilever at the end of the experiment (Fig. S3).

We hypothesized that the bacteria might have been killed or their recovery delayed. Delay in post-treatment recovery after an antibiotic is removed from the extracellular environment is known as the postantibiotic effect (PAE), and it impacts antibiotic dosing (16, 17). Colony counts after treatment indicated that AZI killed less than 1 log of *S. enterica* when plates were incubated at 37°C during recovery. However, during RT recovery, the same concentrations of AZI killed up to 2 logs more irrespective of the treatment temperature (Fig. 1D). The enhanced post-treatment killing by AZI at a lower temperature may reflect slower dissociation of the drug from the ribosome, which is known to increase the bactericidal activity of macrolides (18).

The slope of the nanomotion variance during drug exposure is a proxy for estimating drug susceptibility (11). To test whether detection of AZI resistance in *Salmonella* is feasible with nanomotion, we determined the slopes of the variance at different AZI concentrations for wt SL1344 and a resistant mutant *acrB* R717Q, which harbors a clinically relevant mutation that increases AZI efflux in the acrAB-TolC efflux pump and has an MIC of 32 µg/mL (5, 19). We also determined the slope of the variance for wt strain at acidic pH, which increases AZI's MIC above 1,024 µg/mL (Table S1) (15). We used the rolling regression method for slope estimation, which demonstrated better reliability and

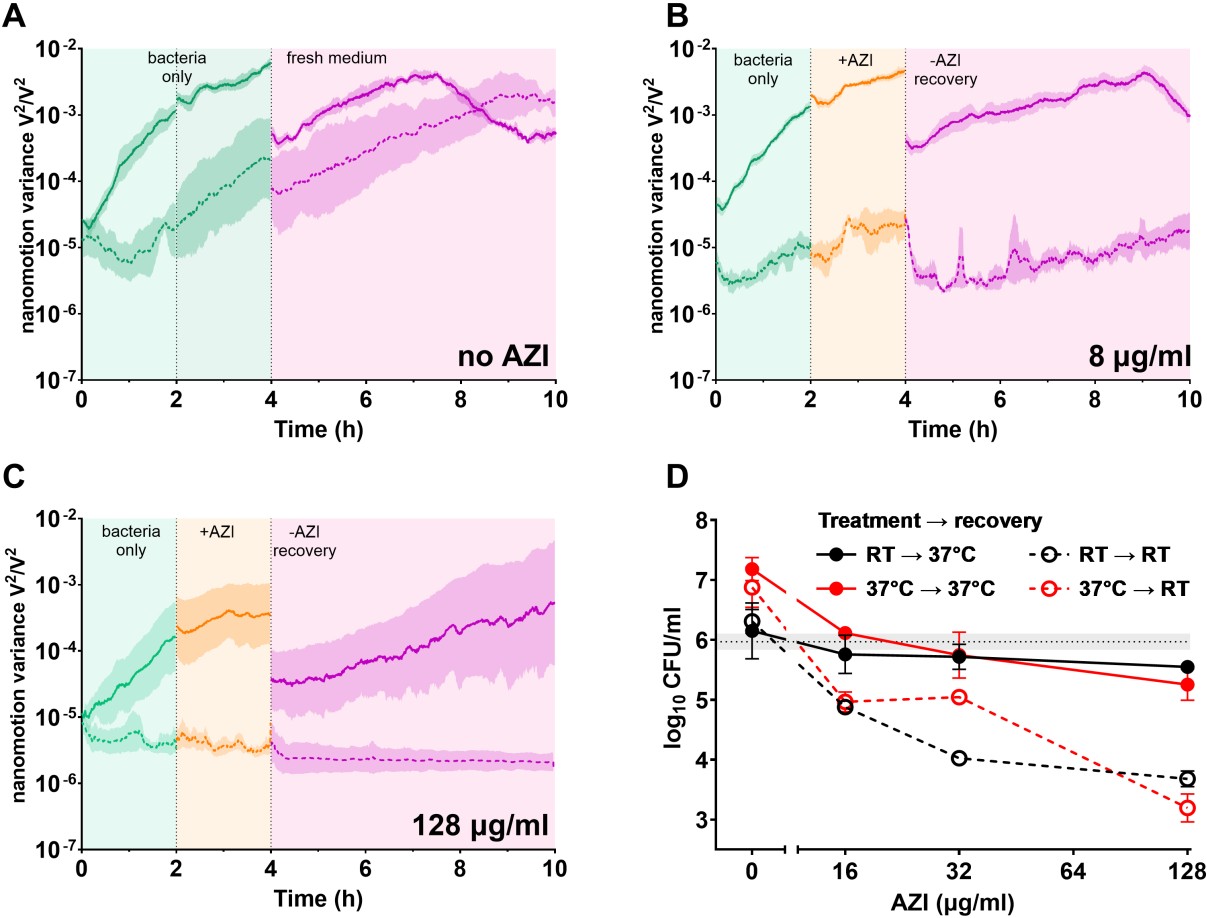

**FIG 1** Variance over time of the nanomotion signal measurements of wt *Salmonella* without AZI (A), with 8 µg/mL (B) or 128 µg/mL (C) AZI for 2 h and subsequent recovery in fresh medium at room temperature (dashed line) or at 37°C (continuous line) at pH 7.4. Green indicates bacterial nanomotion variance signal before adding the drug, orange is during drug treatment, and pink is the recovery in fresh medium after removing the drug. Means ± SEM ($N \geq 3$ biological replicates) shown for nanomotion data. (**D**) Recovery of wt *Salmonella* colonies on lysogeny broth (LB) agar after 2 h of treatment with AZI at indicated temperature at pH 7.4. Gray dotted line indicates the initial inoculum. Means ± SD ($N \geq 3$ biological replicates).

robustness compared to the methods employed in previous studies (Materials and Methods, Fig. S4).

Remarkable differences in nanomotion arise between the strains at AZI concentrations near the MIC value of the wt at neutral pH. The drug slope values of the resistant mutant begin decreasing at higher AZI concentrations than the wt (Fig. 2A). A comparable difference is seen in the wt between neutral and acidic pH (Fig. 2B), indicating that nanomotion can be used to detect AZI susceptibility.

Drug slopes started to decline at AZI concentrations severalfold below the MIC, indicating an effect on the bacteria (Fig. 2). To verify these sub-MIC effects of AZI, we used a fluorescent reporter in which the translational attenuation-based regulatory leader region (*ermCL*) is fused to green fluorescent protein (GFP) (Fig. 2C) instead of the native ermC methyltransferase that confers macrolide resistance (20, 21). Macrolides stall the ribosome during ErmCL translation, which opens the mRNA secondary structure, allowing translation initiation of the downstream gene (21). AZI induced GFP expression in bacteria containing the reporter plasmid in a concentration-dependent manner (Fig. 2D) at these same sub-MIC concentrations where drug slopes began to decline. Maximum reporter induction was observed at or slightly above the MIC at pH 7.4; however, little to no induction was seen at concentrations ≤1 µg/mL (Fig. 2D, Fig. S5 and S6), which is in good agreement with the nanomotion data (Fig. 2A and B). In accordance with the lower MIC at RT, the signal peaked at 4× lower concentrations at RT than it did at 37°C

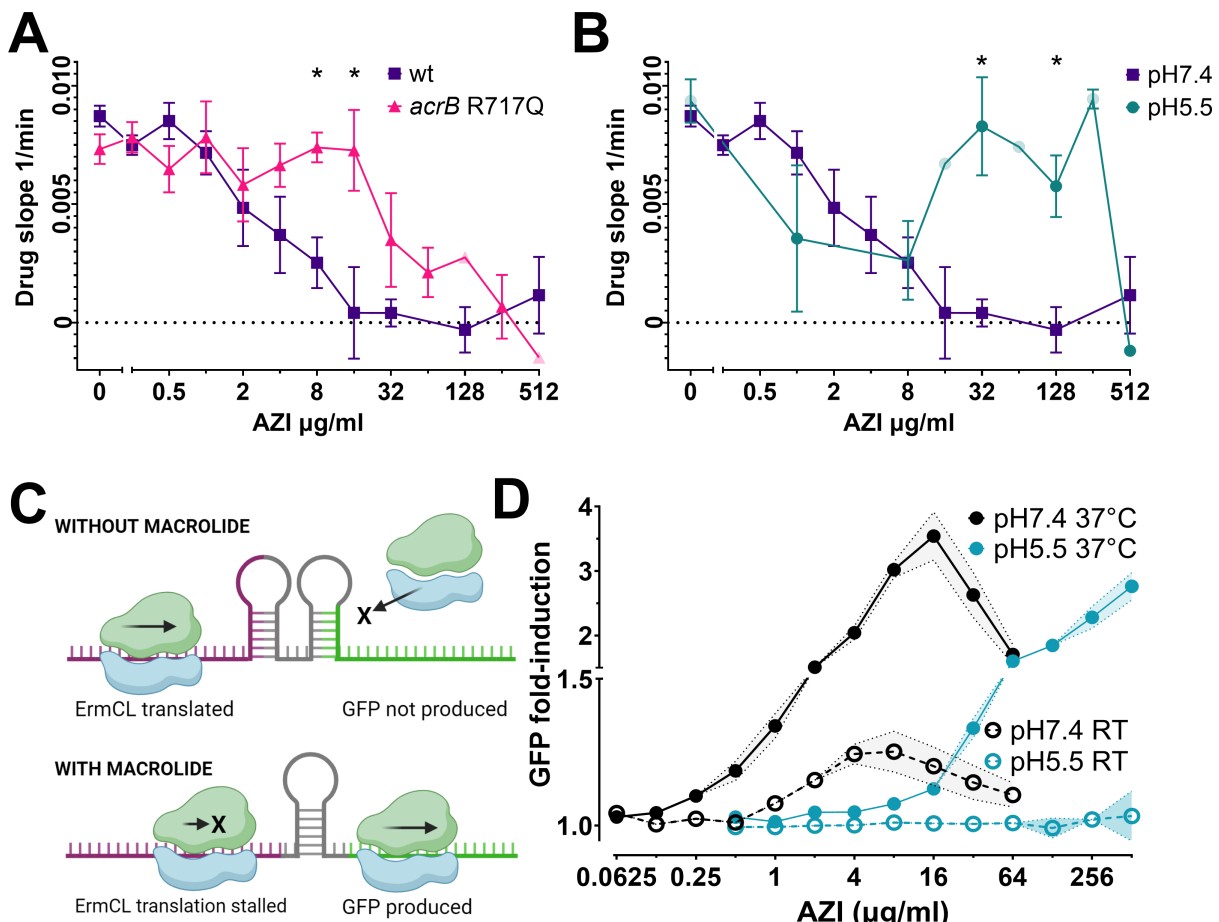

**FIG 2** Effect of AZI on resistant mutants of *Salmonella* and wild type at different pH. (A) The slope of the variance during the drug phase of the AZI-resistant acrB R717Q mutant and susceptible wt strain SL1344 at neutral pH. (B) Drug slope of the wt strain at two different pH values. RT data; means ± SEM (*N* ≥ 3); transparent data points shown, where *N* < 3. * indicates *P*-value <0.05 of the difference between the groups at the indicated concentration. (C) Schematic representation of the ErmCL-based reporter system. (D) Green fluorescent protein (GFP) induction of the ErmCL reporter in wt *Salmonella* after 2 h treatment with AZI. Flow cytometry data. Means ± SD (*N* = 3).

(Fig. 2D; Fig. S6). However, the induction levels remained significantly lower at RT, reflecting slower translation processes. At pH 5.5, GFP induction began at significantly higher concentrations compared to pH 7.4 (Fig. 2D, Fig. S5 and S6), supporting the notion that the pH dependence of AZI sensitivity is due to differences in antibiotic accumulation within the cell. GFP signal was not detected at RT at pH 5.5 (Fig. 2D; Fig.S6).

In summary, we show that nanomotion technology can be used for rapid detection of AZI susceptibility. MIC values obtained using the standard dilution method, CFU counting results, and ermCL-dependent GFP induction by AZI were all consistent with the physiological responses recorded by nanomotion. Additionally, we found that nanomotion is effective for detecting PAE and assessing bactericidal activity. Our study highlights the importance of assay conditions, which significantly affected AZI efficacy. Considering this, increased susceptibility *in vivo*, compared to artificial standard AST laboratory conditions, might also explain AZI's efficacy during the treatment of *Salmonella* infections despite low serum concentrations.

*Salmonella enterica* serovar Typhimurium SL1344 was used as the wt, and the same strain with *acrB* R717Q mutation conferring resistance to azithromycin was used as the resistant strain (5, 15). Both strains carry streptomycin resistance for selection. Bacteria were streaked from −80°C glycerol stocks onto lysogeny broth (LB) agar (BD Difco™ LB agar, Lennox) containing 90 µg/mL streptomycin (Strep90), incubated at 37°C, and the plates were stored at +4°C up to 1 week.

A total of 25 mg/mL stock solution of AZI (Carbosynth, UK) was prepared in 96% ethanol and stored at +4°C. Also, a total of 25 mg/mL stock solution of inactive chloramphenicol (CAM) (Calbiochem, Merck, Germany), likely an inactive stereoisomer, was prepared in 100% ethanol and stored at −20°C. The lack of antibiotic activity on bacterial culture was confirmed via disk diffusion assay (Fig. S7). Carbenicillin was used for pZb-ErmCL-GFP selection at 100 µg/mL.

Stock solutions (1 M) of 4-(2-hydroxyethyl)-1-piperazineethanesulfonic acid (HEPES) (Sigma, USA; Fisher Scientific, Taiwan) or β-(N-morpholino)ethanesulfonic acid (MES) (AppliChem, Germany) were prepared as follows: powder was dissolved in $dH_2O$ by boiling in a microwave, solutions were cooled, and the pH was adjusted to 7.4 (HEPES) or 5.5 (MES) with NaOH. Buffer solutions were filter-sterilized using a 0.22 µm pore size filter. Buffer stocks were stored at +4°C.

Concentrations of 2×, 1.1×, or 1× BBL Mueller Hinton II Broth (Cation-Adjusted) (MHB) (BD, USA) medium were prepared in $dH_2O$, sterilized by autoclaving, and stored at room temperature. Buffered MHB was prepared daily. A 20% Bacto Casamino Acids (CAA) (BD, USA) solution was filter-sterilized using a 0.22 µm pore size filter and stored at +4°C up to 7 days. Low phosphate, low magnesium (LPM) medium (22) stock was prepared in 1.1× concentration and autoclaved. Final concentrations of 0.3% vol/vol glycerol, 0.1% CAA, $MgCl$ (from a 1M stock), and buffer were added to the LPM stock directly prior to the experiment. A 10× stock of 3-(N-morpholino)propanesulfonic acid (MOPS)-based minimal medium (23) was prepared in $dH_2O$, sterile-filtered, and stored at −20°C. Final concentrations of 9.54 mM $NH_4Cl$, 1.32 mM $K_2HPO_4$, 0.2% glycerol or glucose, 0.2% CAA, and buffer were added to stock MOPS medium, filled up to volume with sterile $dH_2O$, and used immediately.

Nanomotion was recorded with Resistell's Phenotech device (Resistell, Switzerland). Resistell technology is described in more detail under patent WO2023174728A1 and in publication (11).

A single colony was used to start a 3 mL overnight culture in MHB with Strep90. On the day of the experiment, the overnight culture was diluted to OD600 0.1 into fresh medium without antibiotics. The day culture was grown aerobically up to OD600 ~ 0.5. Then, 1 mL of culture was centrifuged at 2,000 rcf (ScanSpeed Mini, LABOGENE) for 3 min. The supernatant was removed, and the pellet was resuspended in 160 µL of 1× phosphate-buffered saline (PBS). A 1% agarose solution (low melting point, Gibco, Thermo Fisher Scientific) was melted at 95°C and diluted to 0.2% in 1× PBS. A total of 40 µL was added to the resuspended bacteria, yielding a final concentration of 0.02% agarose. To facilitate attachment, droplets of the bacteria-agarose mixture were carefully applied to a cantilever that had been pre-treated with 0.1 mg/mL poly-D-lysine hydrobromide. The bacteria were allowed to attach for up to 3 min. The functionalization of the cantilever is described in detail in a previous study (11). After attachment, the cantilever was gently washed with 1× PBS, and attachment quality was assessed via phase contrast microscopy (EVOS XL, Thermo Fisher Scientific). In case bacterial attachment was unsatisfactory, the procedure was repeated for an additional 2 min until bacterial coverage was obtained.

In the Phenotech device, first, a 10 min nanomotion blank phase was recorded to get a background signal from the chamber containing 2 mL of growth medium (MHB containing 100 mM HEPES or MES buffer) and the cantilever before attachment of bacteria. Then, after the attachment procedure, the cantilever was reinserted into the measurement chamber, and a bacterial signal was recorded for 2 h. This time also allowed the bacteria to adjust to the new environment. For the drug phase, AZI solution (i.e., 5 to 40 µL solution in ethanol) was added directly to the medium in the chamber and gently mixed, and recording resumed for an additional 2 h. In AZI-free samples, ethanol solution, inactive CAM, or nothing was added to the chamber. These AZI-free samples showed similar nanomotion and were analyzed together to increase sample size. For recovery experiments, the medium was gently removed from the chamber with a pipette and replaced with 2 mL of fresh medium at the same temperature. For

experiments conducted at 37°C, the measurement head of the Phenotech device was mounted in an incubator.

Each sampled nanomotion signal was split into 10 second timeframes. For each timeframe, the linear trend was removed, and the variance of the residual frame was estimated. An additional smoothing procedure was applied using a running median with a 1 min time window and a stride of one element (16.66 seconds) to smooth the variance signal, facilitating plot interpretation.

To calculate the slope of the variance in the drug phase for determining the nanomotion dose response, we used the formula $\log(x) = \log(C) + at$, where $t$ is time (in minutes), $a$ is the slope of the common logarithm of the variance trend, and $\log(C)$ is the intercept. Variance plots were employed for visual inspection of results and currently serve as the primary tool for investigators.

To enhance the robustness of the variance analysis, we used an alternative approach called rolling regression, where the estimate is the median of the slopes obtained from this method. A sliding window of 1,000 seconds (100 data points) with a stride of 200 seconds (20 data points) was applied to the variance signal, which had a 10 second sampling period. Within each window, linear regression was performed to extract the slope of the local curve. The distribution of slopes from these overlapping windows was then summarized using the median as the descriptive statistic. The resulting graphs are shown in Fig. S4.

All final figures were created using GraphPad software version 10.2.2. For statistical analysis, an unpaired $t$-test was performed assuming individual variance for each row. $P$-value threshold was set to 0.05. No correction for multiple comparisons was made.

pZb-ErmCL-GFP (isopropyl β-d-1-thiogalactopyranoside [IPTG]-inducible, ampicillin-resistant) was a gift from Alexander Mankin, University of Illinois Chicago. The plasmid is similar to the plasmid pErmCL-RFP (20), but instead of the red fluorescent protein (RFP), it has GFP as the fluorescent reporter protein. The plasmid was introduced into *Salmonella* wt cells via electroporation and selected on LB agar plates containing 100 µg/mL carbenicillin (Cb100) and 90 µg/mL streptomycin (Strep90). A 15% glycerol stock was prepared from an overnight aerobic liquid culture of a single colony and stored at −80°C. Cells were streaked from the glycerol stock onto selective LB agar medium, and a single colony was used for making an overnight aerobic liquid culture in MHB (with Cb100/Strep90), and 8% dimethyl sulfoxide (DMSO) stocks were prepared, aliquoted, and stored at −80°C.

To prepare 1M IPTG (Apollo Scientific, UK), it was dissolved in deionized water (dH$_2$O) and filter-sterilized. The solution was stored at −20°C.

A culture was started from 120 µL DMSO stock added to 9 mL of 1.1× MHB medium in a 100 mL flask and incubated aerobically for 1 h. A total of 4.5 mL of the culture was transferred to a new 100 mL flask, and 500 µL of 1M buffer (HEPES at pH 7.4 or MES at pH 5.5) as well as 5 µL of 1M IPTG was added to each flask. Flasks were incubated for an additional 1 h. A total of 100 µL of culture adjusted to OD600 0.2 was added to 100 µL of serially diluted azithromycin on a microtiter plate in the same medium, yielding a starting OD600 of 0.1. Plates were incubated in the BiotTek Synergy Mx plate reader with continuous shaking for 2 h at 37°C or at 25°C as room temperature. OD600 and GFP fluorescence (Ex: 479/9,0, Em: 520/20,0, Gain: 100) were recorded every 10 min. Data were normalized according to OD600, and GFP fold difference from the non-antibiotic control was calculated as the induction level (Fig. S6). Plates were chilled on ice, and dilutions were made into sterile-filtered 1× PBS for flow cytometry analysis (Attune NxT Acoustic Focusing Cytometer). Blue laser (488 nm) and emission filter 530/30 nm were used for GFP detection. All incubations were done at 37°C, unless stated otherwise. Data were analyzed with FlowJo software. Bacterial cells were gated from flow cytometry noise according to side scatter (height) and green fluorescence (height) (Fig. S8). To calculate reporter induction, geometric means of green fluorescence (Fig. S5) were used to calculate fold change of treated to non-treated samples.

MICs were determined according to standard protocol via microdilution assay (24). Shortly, overnight cultures in MHB (with Strep90) were diluted into fresh MHB without antibiotics and grown aerobically at 37°C until OD600 reached ~0.5. Cells were diluted to initial inoculum $5 \times 10^5$ CFU/mL into serially diluted antibiotic solution in growth medium. The plates were then incubated at the indicated temperature for about 20 h, and the MICs were determined visually. The results are summarized in Table S1.

During nanomotion experiments, the resistance phenotypes were randomly checked by disk diffusion assay for internal control. For this, exponential bacterial cultures (OD600 ~0.5) used for attachment were spread on MHB agar plates with sterile cotton swabs, and a filter disk containing antibiotic at a dose recommended by The European Committee on Antimicrobial Susceptibility Testing, i.e., 15 µg for AZI, 30 µg for CAM (25), was placed on the agar plate. The inhibition zones were measured after ~18 h of incubation at 37°C and compared with breakpoint tables. Inhibition zones of all tested cultures were in accordance with the expected resistance phenotype.

A total of 3 mL day cultures in MHB was started from overnight cultures and grown aerobically to OD600 ~0.5. At the start of the experiment, cultures were diluted to OD600 of 0.1 in 1× PBS, and diluted further into indicated media without antibiotics to yield a $5 \times 10^5$ CFU/mL in 200 µL volume at the start of the experiment. The cultures were incubated statically at 37°C. Samples were taken every hour until 3 h, mixed in equal volumes with 30% glycerol in 1× PBS, and stored at −80°C. For cell counting, the frozen samples were thawed on ice and stained with 5 µM final concentration Syto9 green fluorescent nucleic acid stain (Invitrogen, Thermo Fisher Scientific, USA) for 30 min in the dark at RT. Cells were counted with flow cytometry (Ex: 488 nm; Em: 530/30 nm). Additional dilutions were made before analysis, when necessary. Green fluorescent cells were counted, and the growth rate between 1 and 3 h was calculated as generation time in minutes.

Survival after 2 h treatment with AZI was determined similarly to the MIC protocol, with a major difference being a two times higher initial inoculum and subsequent CFU plating. The day cultures of wt strain were started from DMSO stocks and grown aerobically in 3 mL MHB medium without antibiotics at 37°C until OD600 ~0.5. AZI was serially diluted on a microtiter plate in MHB medium buffered with 100 mM HEPES at pH 7.4. At the start of the experiment, the bacterial culture was diluted to ~$1 \times 10^6$ CFU/mL in 100 µL final volume of antibiotic-containing medium. The plates were incubated statically at RT (23°C) or 37°C for 2 h. After treatment, the cultures were thoroughly mixed by pipetting, and dilutions were made in 1 mL volumes in 1× PBS. For recovery at RT, the cells were washed once with 1× PBS before making dilutions, as preliminary experiments revealed growth inhibition on CFU plates due to AZI carryover in less dilute samples, which was not a problem when CFU plates were incubated at 37°C. Between 50 and 100 µL of the dilution was plated on LB agar plate, plates were incubated at 37°C or at RT (23°C–24°C), and CFUs were counted the next day (37°C) or after 2 days (RT) of incubation.

## ACKNOWLEDGMENTS

This research was funded by Estonian Research Council grants PRG2696 and MOB3ERA7 Effective RApid DIagnostics and treatment of AntiMicrobial Resistant bacteria (ERAD-IAMR), and EU TWINNING project "Molecular Infection Biology Estonia–Research Capacity Building" (H2020-WIDESPREAD-2018-2020/GA: 857518).

We are thankful to Dorota Klepacki and Alexander Mankin (University of Illinois at Chicago) for preparing and sharing the reporter plasmid. We are thankful to the whole Resistell team for helping with the nanomotion experiments.

M.H.: investigation, methodology, data curation, formal analysis, visualization, writing-original draft; T.M., I.K., M.P.V.: investigation; G.C.: data curation, formal analysis, software; M.P., D.C., A.S., N.K., T.T.: conceptualization, funding acquisition, project administration, supervision. All authors contributed to the review and editing of the

manuscript. Resistell AG has developed the patented (WO2023174728A1) methodology for nanomotion detection.

The ERADIAMR (Effective RApid DIagnostics and treatment of AntiMicrobial Resistant bacteria) is a European project on antimicrobial resistance part of the JPI-AMR action.

## AUTHOR AFFILIATIONS

[1]Institute of Technology, University of Tartu, Tartu, Estonia
[2]Resistell AG, Muttenz, Basel-Landschaft, Switzerland

## PRESENT ADDRESS

Mariliis Hinnu, Institute of Molecular and Cell Biology, University of Tartu, Tartu, Estonia

## AUTHOR ORCIDs

Mariliis Hinnu  http://orcid.org/0000-0002-6096-0658
Niilo Kaldalu  https://orcid.org/0000-0002-1457-2291
Alexander Sturm  https://orcid.org/0000-0002-3818-0428
Tanel Tenson  http://orcid.org/0000-0002-0260-3601

## FUNDING

| Funder | Grant(s) | Author(s) |
|---|---|---|
| Estonian Research Council | PRG335,MOB3ERA7 | Tanel Tenson |
| European Commission | 857518 | Tanel Tenson |

## AUTHOR CONTRIBUTIONS

Mariliis Hinnu, Data curation, Formal analysis, Investigation, Methodology, Visualization, Writing – original draft, Writing – review and editing | Toomas Mets, Investigation, Writing – review and editing | Ivana Kerkez, Investigation, Writing – review and editing | Marta Putrinš, Conceptualization, Funding acquisition, Project administration, Supervision, Writing – review and editing | Niilo Kaldalu, Conceptualization, Funding acquisition, Project administration, Supervision, Writing – review and editing | Gino Cathomen, Data curation, Formal analysis, software, Writing – review and editing | Marta Pla Verge, Investigation, Writing – review and editing | Danuta Cichocka, Conceptualization, Funding acquisition, Project administration, Supervision, Writing – review and editing | Alexander Sturm, Conceptualization, Funding acquisition, Project administration, Supervision, Writing – review and editing | Tanel Tenson, Conceptualization, Funding acquisition, Project administration, Resources, Supervision, Writing – review and editing.

## DATA AVAILABILITY

Data used for constructing the figures in this study are available in the supplementary shared data file. All nanomotion variance data used in this study are available at https://doi.org/10.5281/zenodo.14930488.

## ADDITIONAL FILES

The following material is available online.

### Supplemental Material

**Supplemental material (Spectrum02385-24-s0001.docx).** Fig. S1 to S8; Table S1.
**Supplemental data (Spectrum02385-24-s0002.xlsx).** Data used for Fig. 1D and 2.

## Open Peer Review

**PEER REVIEW HISTORY (review-history.pdf).** An accounting of the reviewer comments and feedback.

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
