## [Reviewer comments · Microbiology Spectrum]

Microbiology Spectrum

Nanomotion technology for testing azithromycin susceptibility of *Salmonella enterica*

Mariliis Hinnu, Toomas Mets, Ivana Kerkez, Marta Putrinš, Niilo Kaldalu, Gino Cathomen, Marta Pla Verge, Danuta Cichocka, Alexander Sturm, and Tanel Tenson

Corresponding Author(s): Mariliis Hinnu, Tartu Ulikool

Review Timeline:

Submission Date:	September 25, 2024
Editorial Decision:	December 6, 2024
Revision Received:	December 19, 2024
Editorial Decision:	January 28, 2025
Revision Received:	February 27, 2025
Accepted:	April 1, 2025

Editor: Mariola Ferraro

Reviewer(s): Disclosure of reviewer identity is with reference to reviewer comments included in decision letter(s). The following individuals involved in review of your submission have agreed to reveal their identity: Mor N Lurie-Weinberger (Reviewer #1); Ying-Tsong Chen (Reviewer #2); Garrett Ellward (Reviewer #3)

Transaction Report:

DOI: <https://doi.org/10.1128/spectrum.02385-24>

Re: Spectrum02385-24 (Nanomotion technology for testing azithromycin susceptibility of Salmonella enterica)

Dear Dr. Mariliis Hinno:

Thank you for the privilege of reviewing your work. Below you will find my comments, instructions from the Spectrum editorial office, and the reviewer comments.

Revision Guidelines

Sincerely,
Mariola Ferraro
Editor
Microbiology Spectrum

Reviewer #1 (Comments for the Author):

Summary of Key Findings (200-250 words)

The paper describes the use of nanomotion technology for rapid phenotypic testing of Salmonella's susceptibility to azithromycin and detection of bactericidal effect. The paper uses various culture conditions, both during azithromycin treatment and

throughout the recovery period. Azithromycin susceptibility in Salmonella is detectable after only two hour of treatment. Thus, it reflects the quick action of the antibiotic, which they hypothesize might be a contributing factor behind the clinical efficacy of azithromycin for Salmonella treatment. The study concentrates on the critical role of assay conditions, which greatly influenced both azithromycin efficacy and the test results. Finally, the results suggest that nanomotion technology can be used in clinical settings for rapid detection of azithromycin resistance in the future.

Major Concerns (at most 5-6):

- The text is at times not very clear and some editing should take place to make the text easier to read. For example, it would have been better to write that the acidic pH is due to the conditions encountered by intracellular Salmonella in acidic vacuoles when this condition was first mentioned around line 71-73, to present the logic behind the conditions selected for the experiments.
- Importance (Lines 41-51) - the importance of the paper should be more focused and precise. There is a repetition here of things from the abstract that redundant.
- Line 75 - " Salmonella is up to 4 times more sensitive to AZI at RT compared to 37{degree sign}C in different growth media" - is there any previous indications of this in the literature?
- Figure 1 - Are the panels numbered correctly? Because panel B is supposedly 37 degrees, but it says RT on the panel itself. Also, what was the pH level at panel A and B? I am worried that the bacteria in Panel B before treatment seems to act differently than bacteria in Panel A green area. Isn't this surprising? And also, why is the concentration chosen 128, isn't this too high? Panel D shows that RT to RT causes a massive effect. I would have liked to see some reference to this in the paper.
- Line 126 - The pH level of 5.5 - was this used also at any point in the previous analyses? And if not, why not?

Minor Concerns (at most 5-20 in bullet points):

- Though important, I feel that the repetition of the fact that the test can be done in two hours redundant. This appears in the abstract, importance, and in line 65. At least one is redundant.
- Line 42-43 - this is an awkwardly worded sentence, I would recommend it edited.
- Figure S7 - the pictures are not very good, it is hard to see the inhibition.

Reviewer #2 (Comments for the Author):

Comments

Need More Technical Details on Nanomotion Technology

The description of nanomotion technology in the manuscript is too brief, lacking essential technical details.

It is unclear how bacteria are prepared and attached to the cantilever sensor, and how bacterial activity translates into measurable oscillations. Also, the experimental setup, including environmental controls (e.g., temperature, medium composition), and how oscillation signals are processed and analyzed, is not well-described. I suggest the authors to provide a more detailed description of the operational procedure and experimental design for nanomotion measurements. Including a diagram or schematic of the experimental setup would greatly aid reader comprehension.

Uncertainty Regarding the Source of Oscillations

Salmonella enterica is a motile bacterium with flagella. It is unclear whether the nanomotion signals measured in this study primarily reflect bacterial metabolic activity, flagellar motion, or a combination of both. This ambiguity raises questions about how azithromycin's effect is being measured:

Azithromycin could inhibit bacterial metabolism or flagellar activity. Does that affect nanomotion signals? However, the manuscript does not address whether the changes detected are due to the drugs bactericidal action or an inhibition of motility-related oscillations. If flagellar motion significantly contributes to the observed oscillations, it is essential to distinguish this from metabolic activity.

I think the authors could consider conducting additional control experiments using non-motile strains (e.g., fliC/D mutants or naturally non-motile bacteria) to confirm the source of oscillations. If such motility contributes significantly, this should be discussed as a limitation, and its implications for the interpretation of azithromycin effects should be clarified.

Review of the paper: "Nanomotion technology for testing azithromycin susceptibility of *Salmonella enterica* "

Summary of Key Findings (200-250 words)

The paper describes the use of nanomotion technology for rapid phenotypic testing of *Salmonella*'s susceptibility to azithromycin and detection of bactericidal effect. The paper uses various culture conditions, both during azithromycin treatment and throughout the recovery period. Azithromycin susceptibility in *Salmonella* is detectable after only two hour of treatment. Thus, it reflects the quick action of the antibiotic, which they hypothesize might be a contributing factor behind the clinical efficacy of azithromycin for *Salmonella* treatment. The study concentrates on the critical role of assay conditions, which greatly influenced both azithromycin efficacy and the test results. Finally, the results suggest that nanomotion technology can be used in clinical settings for rapid detection of azithromycin resistance in the future.

Major Concerns (at most 5-6):

- The text is at times not very clear and some editing should take place to make the text easier to read. For example, it would have been better to write that the acidic pH is due to the conditions encountered by intracellular *Salmonella* in acidic vacuoles when this condition was first mentioned around line 71-73, to present the logic behind the conditions selected for the experiments.
- Importance (Lines 41-51) – the importance of the paper should be more focused and precise. There is a repetition here of things from the abstract that redundant.
- Line 75 – " *Salmonella* is up to 4 times more sensitive to AZI at RT compared to 37°C in different growth media" – is there any previous indications of this in the literature?
- Figure 1 – Are the panels numbered correctly? Because panel B is supposedly 37 degrees, but it says RT on the panel itself. Also, what was the pH level at panel A and B? I am worried that the bacteria in Panel B before treatment seems to act differently than bacteria in Panel A green area. Isn't this surprising? And also, why is the concentration chosen 128, isn't this too high? Panel D shows that RT to RT causes a massive effect. I would have liked to see some reference to this in the paper.
- Line 126 - The pH level of 5.5 – was this used also at any point in the previous analyses? And if not, why not?

Minor Concerns (at most 5-20 in bullet points):

- Though important, I feel that the repetition of the fact that the test can be done in two hours is redundant. This appears in the abstract, importance, and in line 65. At least one is redundant.
- Line 42-43 – this is an awkwardly worded sentence, I would recommend it edited.
- Figure S7 – the pictures are not very good, it is hard to see the inhibition.

Confidential Comments to the Editor:

This is an interesting paper, but it is not well presented. I would argue that the conditions that the researchers have chosen need to be better presented and the logic behind them better explained. There is some repetition in the text and at other sections it is too short. I would advise editing.

Dear Editor,

Please find the submission of revised manuscript “Nanomotion technology for testing azithromycin susceptibility of *Salmonella enterica*”.

Thank you for the opportunity to improve our article. We have followed your suggestions and improved the article according to the questions and concerns raised by the reviewers. Based on the reviewers' suggestions we have modified the text and figures.

We thank the reviewers for constructive feedback. Please find the point-by-point responses (in purple) below. All references to line numbers are to the file with marked changes.

We hope that the revised version of the manuscript is suitable for publication in the Microbiology Spectrum.

With best regards,

Mariliis Hinno and co-authors

Reviewer #1 (Comments for the Author):

Summary of Key Findings (200-250 words)

The paper describes the use of nanomotion technology for rapid phenotypic testing of *Salmonella*'s susceptibility to azithromycin and detection of bactericidal effect. The paper uses various culture conditions, both during azithromycin treatment and throughout the recovery period. Azithromycin susceptibility in *Salmonella* is detectable after only two hour of treatment. Thus, it reflects the quick action of the antibiotic, which they hypothesize might be a contributing factor behind the clinical efficacy of azithromycin for *Salmonella* treatment. The study concentrates on the critical role of assay conditions, which greatly influenced both azithromycin efficacy and the test results. Finally, the results suggest that nanomotion technology can be used in clinical settings for rapid detection of azithromycin resistance in the future.

Major Concerns (at most 5-6):

- The text is at times not very clear and some editing should take place to make the text easier to read. For example, it would have been better to write that the acidic pH is due to the conditions encountered by intracellular *Salmonella* in acidic vacuoles when this condition was first mentioned around line 71-73, to present the logic behind the conditions selected for the experiments.

We edited the text throughout the manuscript to be more clear (especially lines 77-85). We brought the text “acidic media, a condition encountered by intracellular *Salmonella* in acidic vacuoles (16)” from lines 104-105 to lines 72-73, as suggested by the reviewer.

- Importance (Lines 41-51) - the importance of the paper should be more focused and precise. There is a repetition here of things from the abstract that redundant.

We shortened the importance paragraph and focused on the most important aspects, removing the repetitions.

- Line 75 - " Salmonella is up to 4 times more sensitive to AZI at RT compared to 37{degree sign}C in different growth media" - is there any previous indications of this in the literature?

We have not found any previous literature on azithromycin's temperature effects. However, sub-MIC erythromycin, another macrolide, has shown to induce a significant growth inhibition at lower temperatures in *E. coli* (1).

- Figure 1 - Are the panels numbered correctly? Because panel B is supposedly 37 degrees, but it says RT on the panel itself. Also, what was the pH level at panel A and B? I am worried that the bacteria in Panel B before treatment seems to act differently than bacteria in Panel A green area. Isn't this surprising? And also, why is the concentration chosen 128, isn't this too high? Panel D shows that RT to RT causes a massive effect. I would have liked to see some reference to this in the paper.

We have checked the figure 1 subtitle and it states: "at room temperature (RT) (B) or at 37°C (C)", so it is correct. We did add "at pH 7.4" to the end of the sentence.

Indeed, the curves look different before treatment. This is because in the matter of robust analysis we show means of unnormalized curves. However, individual curves can have different baseline values and have slightly different shape, as the signal depends on the efficiency of attachment, i.e. how many bacteria are attached to the cantilever at the start of the experiment, and the adaptation of bacteria inside the device. Due to technical reasons, the attachment quality of bacteria on the cantilever varies, and occasionally the bacteria detach, which is indicated by the drop in variance signal, and increased resonance signal (not shown nor discussed, but used as internal control). Throughout the manuscript, we have only processed data from experiments with sufficient attachment: the slope before adding the drug is positive and a reasonable amount of bacteria are present on the cantilever at the end of the experiment. Despite the different baselines and curve shapes, the general trends are constant. We plan to incorporate normalization of baseline in further studies.

Concentration 128 ug/ml is indeed high, but showed a consistent difference in recovery at two temperatures.

Line 100 – we changed "during RT recovery, the same concentrations of AZI killed at least one log more" to "up to two log-s more".

- Line 126 - The pH level of 5.5 - was this used also at any point in the previous analyses? And if not, why not?

We carried out a few recovery experiments at this pH, however, as the MIC of AZI is so high at acidic pH (≥ 1024 ug/ml), we did not see any impairment of recovery at the highest tested concentration, and we do not expect to see any killing effects. Of note, AZI's solubility in an aqueous solution is poor (depends on the solid form, but most sources state 0.5-2 mg/ml in a neutral environment, slightly more in acidic

environment (2, 3)), therefore increasing the concentration significantly above the MIC to see a killing effect is not feasible.

Minor Concerns (at most 5-20 in bullet points):

- Though important, I feel that the repetition of the fact that the test can be done in two hours is redundant. This appears in the abstract, importance, and in line 65. At least one is redundant.

We removed the statement from the importance paragraph.

- Line 42-43 - this is an awkwardly worded sentence, I would recommend it edited.

We rephrased the sentence.

- Figure S7 - the pictures are not very good, it is hard to see the inhibition.

We adjusted the contrast for better visibility.

Reviewer #2 (Comments for the Author):

Comments

Need More Technical Details on Nanomotion Technology

The description of nanomotion technology in the manuscript is too brief, lacking essential technical details.

It is unclear how bacteria are prepared and attached to the cantilever sensor, and how bacterial activity translates into measurable oscillations. Also, the experimental setup, including environmental controls (e.g., temperature, medium composition), and how oscillation signals are processed and analyzed, is not well-described. I suggest the authors to provide a more detailed description of the operational procedure and experimental design for nanomotion measurements. Including a diagram or schematic of the experimental setup would greatly aid reader comprehension.

Indeed, the reference to the materials and methods section has somehow disappeared from the main text during previous review and editing process, so we reintroduced the sentence "All materials and methods used in this study are detailed in the supplement." to lines 79-80.

The principle of the nanomotion-based AST has been schematically shown and extensively discussed in previously published articles, references 9-15 in the current manuscript, and more recently published (4). As the current manuscript is published as a short-form article, we did not find it necessary to provide a detailed overview of the previously published methodology. We added a sentence "has been described in detail in the referenced articles" to line 64.

Uncertainty Regarding the Source of Oscillations

Salmonella enterica is a motile bacterium with flagella. It is unclear whether the nanomotion signals measured in this study primarily reflect bacterial metabolic activity, flagellar motion, or a combination of both. This ambiguity raises questions about how azithromycin's effect is being

measured:

Azithromycin could inhibit bacterial metabolism or flagellar activity. Does that affect nanomotion signals? However, the manuscript does not address whether the changes detected are due to the drug's bactericidal action or an inhibition of motility-related oscillations. If flagellar motion significantly contributes to the observed oscillations, it is essential to distinguish this from metabolic activity.

I think the authors could consider conducting additional control experiments using non-motile strains (e.g., *fliC/D* mutants or naturally non-motile bacteria) to confirm the source of oscillations. If such motility contributes significantly, this should be discussed as a limitation, and its implications for the interpretation of azithromycin effects should be clarified.

This indeed is a fascinating subject that has intrigued us. It is currently not known, what exactly causes the nanomotions. We hypothesize that the nanomotion signal reflects a combination of metabolic activity (e.g. ribosomal activity and efflux), and motility. All of which are technically challenging to distinguish without affecting other processes, especially in combination with an antibiotic, which also affects cells in various ways. Notably, AZI affects bacterial cell wall permeability, which complicates the use of fluorescent dyes. On the other hand it is known that AZI itself affects flagellar synthesis and motility (5, 6). We know that the used nanomotion-based AST works on non-motile bacteria, such as *M. tuberculosis* (4, 7), *K. pneumoniae* (4), and other cell types (unpublished). We considered using flagellar- and efflux-deficient mutants in this study, but this was not achievable within the timeframe. However, this is definitely something we would like to investigate in the future. The clear advantage of the nanomotion is the real-time measurement, and while the technology can already be used for rapid AST, we hope that in the future we are able to also get more information about bacterial physiology.

References

1. Cruz-Loya, M., Tekin, E., Kang, T.M., Cardona, N., Lozano-Huntelman, N., Rodriguez-Verdugo, A., Savage, V.M. and Yeh, P.J. (2021) Antibiotics Shift the Temperature Response Curve of *Escherichia coli* Growth. *mSystems*, **6**, 10.1128/msystems.00228-21.
2. Adrjanowicz, K., Zakowiecki, D., Kaminski, K., Hawelek, L., Grzybowska, K., Tarnacka, M., Paluch, M. and Cal, K. (2012) Molecular Dynamics in Supercooled Liquid and Glassy States of Antibiotics: Azithromycin, Clarithromycin and Roxithromycin Studied by Dielectric Spectroscopy. Advantages Given by the Amorphous State. *Mol. Pharmaceutics*, **9**, 1748–1763.
3. Cayman Chemical (2022) Product information: azithromycin.
4. Vocat, A., Luraschi-Eggemann, A., Antoni, C., Cathomen, G., Cichocka, D., Greub, G., Riabova, O., Makarov, V., Opota, O., Mendoza, A., *et al.* (2024) Real-time evaluation of macozinone activity against *Mycobacterium tuberculosis* through bacterial nanomotion analysis. *Antimicrobial Agents and Chemotherapy*, **0**, e01318-24.
5. Bala, A., Kumar, R. and Harjai, K. (2011) Inhibition of quorum sensing in *Pseudomonas aeruginosa* by azithromycin and its effectiveness in urinary tract infections. *Journal of Medical Microbiology*, **60**, 300–306.

6. Matsui,H., Eguchi,M., Ohsumi,K., Nakamura,A., Isshiki,Y., Sekiya,K., Kikuchi,Y., Nagamitsu,T., Masuma,R., Sunazuka,T., *et al.* (2005) Azithromycin Inhibits the Formation of Flagellar Filaments without Suppressing Flagellin Synthesis in Salmonella enterica Serovar Typhimurium. *Antimicrobial Agents and Chemotherapy*, **49**, 3396–3403.
7. Vocat,A., Sturm,A., Józwiak,G., Cathomen,G., Świątkowski,M., Buga,R., Wielgoszewski,G., Cichocka,D., Greub,G. and Opota,O. (2023) Nanomotion technology in combination with machine learning: a new approach for a rapid antibiotic susceptibility test for *Mycobacterium tuberculosis*. *Microbes and Infection*, **25**, 105151.

Re: Spectrum02385-24R1 (Nanomotion technology for testing azithromycin susceptibility of Salmonella enterica)

Dear Dr. Mariliis Hinnu:

Thank you for the privilege of reviewing your work. Below you will find my comments, instructions from the Spectrum editorial office, and the reviewer comments.

Revision Guidelines

Sincerely,
Mariola Ferraro
Editor
Microbiology Spectrum

Reviewer #2 (Comments for the Author):

This work addresses an important issue in antimicrobial resistance diagnostics and demonstrates a method with potential for rapid and effective phenotypic testing.

The manuscript is concise and focused, with technical details appropriately provided in the supplemental materials and

previously published studies. This approach ensures the methodology is clear while maintaining the manuscript's readability and emphasis on its key findings.

While the text effectively highlights the study's results, it could be further enhanced by addressing the broader implications of this research. For instance, including a brief mention of potential applications to other antibiotics, bacterial species, or future clinical uses of this technology would add depth to the discussion. However, it is understood that space constraints might have influenced the current scope of the discussion, and the focus on azithromycin and Salmonella remains justified and appropriate.

There are, however, some minor grammatical and typographical issues that could be corrected to improve readability and polish. I suggest these minor modifications:

L25, '...period significantly...' to '...period, significantly...'

L37, '...settings., and can be used of azithromycin in the future.' to '...settings and its future use with azithromycin.'

L62, susceptibilityof >> susceptibility of

L104, several-fold >> severalfold

L254, at at >> at

Reviewer #3 (Public repository details (Required)):

Submission to ASM journals typically requires data to be submitted alongside the paper. In this case, data for the calculation of variance curves and slope would be appreciated to better support and understand the supposed findings in this article.

Reviewer #3 (Comments for the Author):

Comments For Authors,

The paper "Nanomotion technology for testing azithromycin susceptibility of Salmonella enterica" examines the use of nanomotion detection for azithromycin resistance. In this study, the authors utilize nanomotion to examine the effects of azithromycin treatment on susceptible and resistant isolates of S. enterica and describe environmental conditions that affect azithromycin treatment. I believe that, while this work is promising, additional editing is required before acceptance. This reviewer thanks the authors for their work and presents the following comments for the publishing of this work:

Abstract

1. Abstract falls within 250-word limit. However, I believe the Abstract would be better if additional information concerning the significance of treating S. enterica is added-global burden and need for treatment worldwide.

2. A short description of nanomotion and how it can be used to detect microbial sensitivities should be added.

Importance

1. Line 37. Remove the period and the comma following the word settings.

Main Text

1. Lines 41-42. I failed to find the claim that "all invasive serovars of S. enterica" contain resistance to beta-lactams and fluroquinolones in the cited source. If the authors could provide text for this claim or reword, that would be appreciated.

2. Lines 43-45. The serum concentration listed is correct according to the source, but the cited study also lists that the tissue concentration of azithromycin can remain high and that the antibiotic can still be effective in treatment. If serum concentration is important to S. enterica specifically, the authors should clarify the discrepancy with antibiotic bioactivity in serum.

3. Lines 46-54. While nanomotion technology has been described in previous studies, this reviewer believes this article will benefit from the addition of information concerning how nanomotion technology is used to study bacteria to orient the reader. Including:

How can nanomotion be used to determine resistance or susceptibility?

How is machine learning used to determine susceptibility? Is it compared to standard curves?

3b. Additional information concerning the necessity for room temperature experimentation is required, especially with 37°C being the standard for clinical comparison.

4. Lines 53-54. Add information as to what these clinical studies examined. What did they conclude?
5. Line 62. Add a space between susceptibility and of.
6. Line 63. Add a space between assays and at. Capitalize at.
7. Lines 68-69. The authors demonstrate the nanomotion variances in Figure 1 with azithromycin concentrations at 128 µg/mL (in article), 16 µg/mL (supplement), and 32 µg/mL (supplement) against the wild type *S. enterica* SL1344. If the variances and nanomotion technology can be used to detect sensitive and resistant strains in clinical settings, the authors should also show recorded variances for 8 µg/mL, which is the described MIC concentration in this study. Even with 128 µg/mL, (16 times the listed MIC), variance began to increase approximately two hours after the addition of azithromycin. Though the difference in variance between no antibiotic and antibiotic is great, is this difference maintained at or slightly below listed MIC?
- 7b. Authors should show the variation curve for 37°C and no azithromycin, as that would correspond more to clinical settings.
8. For figure 2, the data suggests the slope of the variance or the drug treatment of the sensitive isolate is near zero (horizontal) at 16 µg/mL. In the supplement, the data seem to suggest the opposite, with the drug treatment variance rising steadily.

Methods

1. Methods and Materials should be added to the paper, as per the requirements of the journal.

This work addresses an important issue in antimicrobial resistance diagnostics and demonstrates a method with potential for rapid and effective phenotypic testing.

The manuscript is concise and focused, with technical details appropriately provided in the supplemental materials and previously published studies. This approach ensures the methodology is clear while maintaining the manuscript's readability and emphasis on its key findings.

While the text effectively highlights the study's results, it could be further enhanced by addressing the broader implications of this research. For instance, including a brief mention of potential applications to other antibiotics, bacterial species, or future clinical uses of this technology would add depth to the discussion. However, it is understood that space constraints might have influenced the current scope of the discussion, and the focus on azithromycin and Salmonella remains justified and appropriate.

There are, however, some minor grammatical and typographical issues that could be corrected to improve readability and polish. There are still some minor typographical and grammatical errors in the manuscript. However, these issues are minor and do not overshadow the overall quality and significance of the work. Correcting these errors will further enhance the clarity and polish of the manuscript:

L25, '...period significantly...' to '...period, significantly...'

L37, '...settings., and can be used of azithromycin in the future.' to '...settings and its future use with azithromycin.'

L62, susceptibilityof >> susceptibility of

L104, several-fold >> severalfold

L254, at at >> at

Comments For Authors,

The paper “Nanomotion technology for testing azithromycin susceptibility of *Salmonella enterica*” examines the use of nanomotion detection for azithromycin resistance. In this study, the authors utilize nanomotion to examine the effects of azithromycin treatment on susceptible and resistant isolates of *S. enterica* and describe environmental conditions that affect azithromycin treatment. I believe that, while this work is promising, additional editing is required before acceptance. This reviewer thanks the authors for their work and presents the following comments for the publishing of this work:

Abstract

1. Abstract falls within 250-word limit. However, I believe the Abstract would be better if additional information concerning the significance of treating *S. enterica* is added—global burden and need for treatment worldwide.

2. A short description of *nanomotion* and how it can be used to detect microbial sensitivities should be added.

Importance

1. Line 37. Remove the period and the comma following the word *settings*.

Main Text

1. Lines 41-42. I failed to find the claim that “all invasive serovars of *S. enterica*” contain resistance to beta-lactams and fluoroquinolones in the cited source. If the authors could provide text for this claim or reword, that would be appreciated.

2. Lines 43-45. The serum concentration listed is correct according to the source, but the cited study also lists that the tissue concentration of azithromycin can remain high and that the antibiotic can still be effective in treatment. If serum concentration is important to *S. enterica* specifically, the authors should clarify the discrepancy with antibiotic bioactivity in serum.

3. Lines 46-54. While nanomotion technology has been described in previous studies, this reviewer believes this article will benefit from the addition of information concerning how nanomotion technology is used to study bacteria to orient the reader. Including:

How can nanomotion be used to determine resistance or susceptibility?

How is machine learning used to determine susceptibility? Is it compared to standard curves?

3b. Additional information concerning the necessity for room temperature experimentation is required, especially with 37°C being the standard for clinical comparison.

4. Lines 53-54. Add information as to what these clinical studies examined. What did they conclude?

5. Line 62. Add a space between *susceptibility* and *of*.

6. Line 63. Add a space between *assays* and *at*. Capitalize *at*.

7. Lines 68-69. The authors demonstrate the nanomotion variances in Figure 1 with azithromycin concentrations at 128 µg/mL (in article), 16 µg/mL (supplement), and 32 µg/mL (supplement) against the wild type *S. enterica* SL1344. If the variances and nanomotion technology can be used to

detect sensitive and resistant strains in clinical settings, the authors should also show recorded variances for 8 µg/mL, which is the described MIC concentration in this study. Even with 128 µg/mL, (16 times the listed MIC), variance began to increase approximately two hours after the addition of azithromycin. Though the difference in variance between no antibiotic and antibiotic is great, is this difference maintained at or slightly below listed MIC?

7b. Authors should show the variation curve for 37°C and no azithromycin, as that would correspond more to clinical settings.

8. For figure 2, the data suggests the slope of the variance or the drug treatment of the sensitive isolate is near zero (horizontal) at 16 µg/mL. In the supplement, the data seem to suggest the opposite, with the drug treatment variance rising steadily.

Methods

1. Methods and Materials should be added to the paper, as per the requirements of the journal.

Dear Editor,

We have revised the manuscript “Nanomotion technology for testing azithromycin susceptibility of *Salmonella enterica*”.

We have followed the reviewers’ suggestions and improved the article by modifying the text and figures (1A to 1C, and S2). Importantly, we have shared all the data according to the open data-sharing policy, and have moved the *Materials and methods* section from the supplementary file to the main manuscript.

We thank the reviewers for their constructive feedback. Please find the point-by-point responses to their concerns (in purple) below. All references to line numbers are to the file with marked changes.

We hope that the revised version of the manuscript is suitable for publication in the *Microbiology Spectrum*.

With best regards,

Mariliis Hinno and co-authors

Reviewer #2 (Comments for the Author):

This work addresses an important issue in antimicrobial resistance diagnostics and demonstrates a method with potential for rapid and effective phenotypic testing.

The manuscript is concise and focused, with technical details appropriately provided in the supplemental materials and previously published studies. This approach ensures the methodology is clear while maintaining the manuscript's readability and emphasis on its key findings.

*While the text effectively highlights the study's results, it could be further enhanced by addressing the broader implications of this research. For instance, including a brief mention of potential applications to other antibiotics, bacterial species, or future clinical uses of this technology would add depth to the discussion. However, it is understood that space constraints might have influenced the current scope of the discussion, and the focus on azithromycin and *Salmonella* remains justified and appropriate.*

We added a sentence to the importance paragraph (lines 37-40): “Additionally, the study shows that nanomotion technology can be used for susceptibility and postantibiotic effect testing for various pathogens and antibacterials, including those generally regarded as bacteriostatic.”

There are, however, some minor grammatical and typographical issues that could be corrected to improve readability and polish. I suggest these minor modifications:

L25, '...period significantly...' to '...period, significantly...'

L37, '...settings., and can be used of azithromycin in the future.' to '...settings and its future use with azithromycin.'

L62, susceptibilityof >> susceptibility of
L104, several-fold >> severalfold
L254, at at >> at

We have corrected the mistakes.

Reviewer #3 (Public repository details (Required)):

Submission to ASM journals typically requires data to be submitted alongside the paper. In this case, data for the calculation of variance curves and slope would be appreciated to better support and understand the supposed findings in this article.

Reviewer #3 (Comments for the Author):

Comments For Authors,

The paper "Nanomotion technology for testing azithromycin susceptibility of Salmonella enterica" examines the use of nanomotion detection for azithromycin resistance. In this study, the authors utilize nanomotion to examine the effects of azithromycin treatment on susceptible and resistant isolates of S. enterica and describe environmental conditions that affect azithromycin treatment. I believe that, while this work is promising, additional editing is required before acceptance. This reviewer thanks the authors for their work and presents the following comments for the publishing of this work:

Abstract

1. Abstract falls within 250-word limit. However, I believe the Abstract would be better if additional information concerning the significance of treating S. enterica is added-global burden and need for treatment worldwide.

We highlighted the significance in the abstract lines 18-22.

2. A short description of nanomotion and how it can be used to detect microbial sensitivities should be added.

Added "which is based on measuring changes in bacterial nanoscale movements" to line 23.

Importance

1. Line 37. Remove the period and the comma following the word settings.

Removed.

Main Text

1. Lines 41-42. I failed to find the claim that "all invasive serovars of S. enterica" contain resistance to

beta-lactams and fluoroquinolones in the cited source. If the authors could provide text for this claim or reword, that would be appreciated.

We reworded “has emerged in all invasive *S. enterica* serovars” to “has been reported in all clinically relevant *S. enterica* serovars”. Hopefully, this clarifies the issue.

2. Lines 43-45. *The serum concentration listed is correct according to the source, but the cited study also lists that the tissue concentration of azithromycin can remain high and that the antibiotic can still be effective in treatment. If serum concentration is important to S. enterica specifically, the authors should clarify the discrepancy with antibiotic bioactivity in serum.*

We agree that tissue concentrations can be vastly different from serum concentrations, and the efficacy of the drug then depends on pathogen localization. However, serum concentration is still one of the most important classical pharmacokinetic parameters, mostly due to simple measurability. According to PK/PD approach to antibiotic dosing, drugs are categorized as dose-dependent (e.g. fluoroquinolones) and time-dependent (e.g. macrolides), in relation to the standard MIC (1). Serum concentrations are taken into account when determining the clinical usefulness of the antibiotic, e.g. (2). It can be seen from EUCAST MIC distribution table for azithromycin that epidemiologic cut-off value (i.e. used for recommending a clinical breakpoint) has been given only when the majority of tested isolates have an MIC near or below the maximum serum concentration (0.5 ug/ml), with a couple of exceptions, incl. *Salmonella* (3). It is very difficult to measure antibiotic concentrations encountered by bacteria in tissues *in vivo*. While high tissue concentrations can explain azithromycin’s efficacy against *Salmonella in vivo*, it is also possible that *Salmonella’s in vivo* effective AZI concentration is much lower than the standard laboratory-determined MIC. Our study highlights that *Salmonella* susceptibility can significantly differ depending on environmental factors, such as pH and temperature, both during and after treatment. Therefore, it is not possible to draw a straight line between *in vitro* susceptibility and serum concentration. We added the last sentence (lines 136-138): “Considering this, increased susceptibility *in vivo*, compared to artificial laboratory standard AST conditions, might also explain AZI’s efficacy during the treatment of *Salmonella* infections despite low serum concentrations.”

3. Lines 46-54. *While nanomotion technology has been described in previous studies, this reviewer believes this article will benefit from the addition of information concerning how nanomotion technology is used to study bacteria to orient the reader. Including:*

How can nanomotion be used to determine resistance or susceptibility?

We added short descriptions to the introductory section. Lines 57-58 “Effective drug concentrations reduce or stop the nanomotion”.

How is machine learning used to determine susceptibility? Is it compared to standard curves?

Added to lines 59-61: “The ML model is trained on a large set of clinical isolates, based on the standard MIC and nanomotion response at different drug concentrations.”

3b. *Additional information concerning the necessity for room temperature experimentation is required, especially with 37°C being the standard for clinical comparison.*

It is already mentioned in the manuscript lines 71-73 that “In the early stages of development,

nanomotion was measured at ambient room temperature (RT). The current setup uses 37°C for all ASTs to mimic physiological conditions in humans and to decrease the time to results (4).” Indeed, we agree that 37C is more clinically relevant. However, the nanomotion equipment is in constant development by the startup company Resistell, and, as mentioned, in the early stages of development the measurement chamber did not feature a thermostat for temperature control. For 37C experiments the device was placed into an incubator, which was only possible for a limited number of devices at the time of the study. All current models feature a thermostat in the measurement chamber.

The actual susceptibility determination method uses machine learning by comparing standard MIC based on microdilution assay at 37C and nanomotion response for each individual isolate (N > 50), so the models will consider any temperature effects. The temperature effects on bacterial susceptibility were less significant in the previous studies on beta-lactams and fluoroquinolones. The current study was a pilot study on *Salmonella* and azithromycin to prove that we can indeed see a difference between resistant and susceptible conditions by nanomotion. As we saw a significant temperature effect, we elaborated more on this topic. For clinical purposes, ML models have to be trained on much larger datasets, which will be done on the updated devices at 37C.

4. Lines 53-54. Add information as to what these clinical studies examined. What did they conclude?

We added information about the clinical studies: lines 64-66 “for determining antibiotic susceptibility (beta-lactams and fluoroquinolones) of Gram-negative bacteria causing bacteremia and/or sepsis with >90% accuracy.”. Part of the PHENOTECH-1 study has been published in the previous article (4), NANO-RAST is under review in *Clinical Microbiology and Infection*.

5. Line 62. Add a space between susceptibility and of.

Added.

6. Line 63. Add a space between assays and at. Capitalize at.
Removed the full stop.

7. Lines 68-69. The authors demonstrate the nanomotion variances in Figure 1 with azithromycin concentrations at 128 µg/mL (in article), 16 µg/mL (supplement), and 32 µg/mL (supplement) against the wild type *S. enterica* SL1344. If the variances and nanomotion technology can be used to detect sensitive and resistant strains in clinical settings, the authors should also show recorded variances for 8 µg/mL, which is the described MIC concentration in this study. Even with 128 µg/mL, (16 times the listed MIC), variance began to increase approximately two hours after the addition of azithromycin. Though the difference in variance between no antibiotic and antibiotic is great, is this difference maintained at or slightly below listed MIC?

We added a new panel to Fig1B with 8 ug/ml azithromycin at both temperatures. There is only a slight effect on the drug slope (orange) at 37C and a biphasic drug slope at RT, so there is a difference in nanomotion between the two conditions. As azithromycin is considered a bacteriostatic drug, a recovery after short-term (2 h) treatment at near-MICs is expected. When cells are recovered at 37C, the recovery happens even at 128 ug/ml, as about half of the cell population survives (Fig. 1D). No recovery

from azithromycin treatment is seen with nanomotion in conditions when >99% of the cell population is killed. Our nanomotion recovery correlates well with the survival assay by CFU counting.

7b. Authors should show the variation curve for 37°C and no azithromycin, as that would correspond more to clinical settings.

We modified Fig. 1 and added the variation curve at 37C without drugs, with respective corrections in the figure subtitle and references in main text.

8. For figure 2, the data suggests the slope of the variance or the drug treatment of the sensitive isolate is near zero (horizontal) at 16 µg/mL. In the supplement, the data seem to suggest the opposite, with the drug treatment variance rising steadily.

The supplementary figure (in previous version) for 16 ug/ml shows data from a single recovery experiment. The nanomotion drug phase seems to follow a biphasic curve (flattens in the second hour). In Fig2AB we used the rolling regression method (explained graphically in Fig S4) to calculate the slopes, so the effects of ±peaks or biphasicity lessen. In Fig. 2A the error bar is quite wide at 16 ug/ml, as probably the variance near the MIC depends on the exact amount of cells on the cantilever. We have now added data from additional experiments to Fig S2.

Methods

1. Methods and Materials should be added to the paper, as per the requirements of the journal.

We relocated the Materials and Methods section into the main manuscript. The fit the limit of 25 references in total, some references were removed and rearranged.

Re: Spectrum02385-24R2 (Nanomotion technology for testing azithromycin susceptibility of Salmonella enterica)

Dear Dr. Mariliis Hinnu:

Your manuscript has been accepted, and I am forwarding it to the ASM production staff for publication. Your paper will first be checked to make sure all elements meet the technical requirements. ASM staff will contact you if anything needs to be revised before copyediting and production can begin. Otherwise, you will be notified when your proofs are ready to be viewed.

Sincerely,
Mariola Ferraro
Editor
Microbiology Spectrum

Reviewer #2 (Comments for the Author):

The manuscript has been substantially improved and now presents a clear and convincing case for the use of nanomotion-based approaches in testing Salmonella resistance to azithromycin. The experimental design is appropriate, the data interpretation is sound, and the conclusions are supported by the results.

Reviewer #3 (Comments for the Author):

This reviewer would like to thank the authors for their work in addressing previous comments. The addition of text as well as data has strengthened the claims made in the study. However, I would like to address potential minor text errors:

Line 91: Potential typo with the inclusion of the hyphen in "log-s"

Line 134: Italicize "enterica"

Lines 165 & 179: I am unsure if the headers "Attachment" and "Nanomotion recording..." are intended to fall under the previous heading or are intended to be included as sub-section. Please confirm editing in this regard.

Lines 393 and 401: Italicize "Salmonella"

This reviewer would like to thank the authors for their work in addressing previous comments. The addition of text as well as data has strengthened the claims made in the study. However, I would like to address potential minor text errors:

Line 91: Potential typo with the inclusion of the hyphen in "log-s"

Line 134: Italicize "enterica"

Lines 165 & 179: I am unsure if the headers "Attachment" and "Nanomotion recording..." are intended to fall under the previous heading or are intended to be included as sub-section. Please confirm editing in this regard.

Lines 393 and 401: Italicize "Salmonella"